# SARS-CoV-2 Antibodies in Breastmilk Three and Six Months Postpartum in Relation to the Trimester of Maternal SARS-CoV-2 Infection—An Exploratory Study

**DOI:** 10.3390/ijms25063269

**Published:** 2024-03-13

**Authors:** Line Fich, Ann-Marie Hellerung Christiansen, Anna Christine Nilsson, Johanna Lindman, Helle Gybel Juul-Larsen, Christine Bo Hansen, Nina la Cour Freiesleben, Mohammed Rohi Khalil, Henriette Svarre Nielsen

**Affiliations:** 1Department of Obstetrics and Gynecology, Copenhagen University Hospital Hvidovre, 2650 Hvidovre, Denmarkhenriette.svarre.nielsen@regionh.dk (H.S.N.); 2Department of Clinical Immunology, Odense University Hospital, 5000 Odense, Denmark; 3Department of Clinical Research, Copenhagen University Hospital Hvidovre, 2650 Hvidovre, Denmark; 4Department of Obstetrics and Gynecology, The Fertility Clinic, Copenhagen University Hospital Hvidovre, 2650 Hvidovre, Denmark; 5Department of Clinical Medicine, Faculty of Health and Medical Sciences, University of Copenhagen, 2100 Copenhagen, Denmark; 6Department of Obstetrics and Gynecology, Lillebaelt Hospital, 6000 Kolding, Denmark; mohammed.khalil@rsyd.dk

**Keywords:** SARS-CoV-2, breastmilk, antibodies, maternal–neonate immunology, trimester of infection, human research

## Abstract

The immune system of neonates is immature and therefore knowledge of possible early-life protection against SARS-CoV-2 infection, such as breastfeeding, is of great importance. Few studies have investigated the presence and duration of SARS-CoV-2 antibodies in breastmilk in relation to the trimester of maternal infection during pregnancy, and none with successful participation from all three trimesters. This study has dual objectives (1) in relation to the trimester of infection to examine the frequency, concentration and duration of IgA and IgG antibodies in breastmilk and blood serum in the third and sixth month post-partum in former SARS-CoV-2-infected mothers and (2) to examine the association in pediatric emergency admission of children within the first six months of life compared to children of non-SARS-CoV-2-infected women. The first objective is based on a prospective cohort and the second is based on a nested case–control design. The study participants are women with a former SARS-CoV-2 infection during pregnancy, whose serology IgG tests at delivery were still positive. Maternal blood and breastmilk samples were collected at three and six months postpartum. Serum IgA frequency three months pp was 72.7% (50%, 90% and 60% in the first, second and third trimester) and 82% six months pp (67%, 91% and 82% in the first, second and third trimester). Breastmilk IgA frequency three months pp was 27% (16.6%, 36% and 20% in first, second and third trimester) and 28% six months pp (0%, 38% and 28% in the first, second and third trimester). The highest IgA concentration in breastmilk was found six months post-partum with infection in the third trimester. Serum IgA was detectable more than 400 days post infection, and serum IgG above threshold was found 430 days after date of infection. We found no correlation between serum IgA and breastmilk IgA, nor between serum IgG and breastmilk IgA regardless of the trimester of infection.

## 1. Introduction

Due to the immature immune system of neonates and the hesitation to vaccinate them against SARS-CoV-2, knowledge of possible early-life protection against SARS-CoV-2 infection, such as breastfeeding, is of great importance [1,2,3]. In a systematic review, Low et al. found 82.6% (133/161) of available samples from 14 studies to have a presence of SARS-CoV-2 IgA or IgG antibodies and 41.7% (20/48) of samples to have SARS-CoV-2 neutralizing antibodies present. Young et al. demonstrated, in their observational cohort study, that infection was associated with a robust SARS-CoV-2 antibody response in breastmilk [4,5]. As observed in Fich et al., the trimester of maternal infection can influence the level of SARS-CoV-2 antibodies transferred to the fetus, and therefore the trimester of SARS-CoV-2 infection could be relevant for the development and duration of antibodies in breastmilk [6]. In a study by Wachman et al., they found a higher level of breastmilk IgA at delivery after early-gestation infections compared with late-gestation infection. Both Wachman et al. and Bobak et al. included participants from various trimesters of infection, and reported their results divided into early and late gestational age, and, therefore, lack presentation in relation to all three trimesters of infection. The study by Szcygiol and colleagues confirms the presence of SARS-CoV-2 IgA and IgG antibodies in the breastmilk of SARS-CoV-2--recovered women and found no relation of the antibody levels and the trimester of infection. The study, although including all three trimesters, does not elaborate on their specific findings in relation to the trimester of infection, and only collected breastmilk samples at one timepoint [7,8,9]. Based on the findings of Fich et al. and the lack of representation of all three trimesters in the literature, we hypothesize that the timing of infection in naturally immunized mothers could have a possible impact on future SARS-CoV-2 maternal–neonate immunization. This study has dual objectives: (1) in relation to the trimester of infection to examine the frequency, concentration and duration of IgA and IgG antibodies in breastmilk and blood serum in the third and sixth month post-partum in former SARS-CoV-2-infected mothers and (2) to examine the association in the pediatric emergency admission of children within the first six months of life compared to children of non-SARS-CoV-2-infected women.

## 2. Results

### 2.1. Demographics

Out of the 40 women included, 34 had an IgG antibody test ≥10.00 AU/mL at delivery and six had an IgG antibody test ≤10.00 AU/mL, but all their newborns had an IgG antibody test ≥10.00 AU/mL at the time of delivery. Of these 40 women, only 36 were able to provide breastmilk and serum samples during the third and/or sixth month after delivery. Twenty-two (61.1%) breastmilk samples were collected during the third month and 29 (80.6%) samples were collected during the sixth month. Fifteen women (41.7%) were able to provide samples in both the third and sixth month after delivery. Of the 36 participants, 6 women were infected in the first trimester (16.67%), 11 in the second trimester (30.56%) and 19 in the third trimester (52.78%) (Table 1). Demographic details for both cases and controls are presented in Table 2.

### 2.2. IgA and IgG Frequency in Breastmilk and Serum Three and Six Months Post-Partum

#### 2.2.1. Breastmilk

In total, 27% (6/22) of the available breastmilk samples from the third month post-partum (pp) had IgA levels above the threshold. In relation to the trimester of infection, 16.6% (1/6) of women infected in the first trimester, 36% (4/11) of women infected in the second trimester and 20% (1/5) of women infected in the third trimester had IgA levels above the threshold in their breastmilk samples. 

In total, 28% (8/29) of the available breastmilk samples from the sixth month pp, had detectable IgA. In relation to the trimester of infection, 0% (0/3) of women infected in the first trimester, 38% (3/8) of women infected in the second trimester and 28% (5/18) of women infected in the third trimester had IgA levels above the threshold in their breastmilk samples (Table 3).

In total, 27.7% (10/36) of breastmilk samples available both three and six months pp had IgA levels above the threshold, and of these samples one was infected in the first trimester, four in the second trimester and five in the third trimester of pregnancy. All breastmilk samples (36/36) had detectable IgA three and/or six months post-partum. No IgG antibodies were detected in any of the breastmilk samples. 

#### 2.2.2. Serum

In total, 77% (17/22) of the available serum samples from the third month pp. had IgG levels above the threshold and 72.7% (16/22) had IgA levels above the threshold. In relation to trimester of infection, 67% (4/6) of women infected in the first trimester had IgG levels above the threshold and 50% (3/6) had IgA levels above threshold. A total of 73% (8/11) of women infected in the second trimester had IgG levels above threshold and 90% (10/11) had IgA levels above threshold. All (5/5) of the women infected in the third trimester had IgG levels above threshold and 60% (3/5) had IgA levels above threshold. 

In total, 79% (26/33) of the available serum samples from the sixth month pp. had IgG levels above the threshold and 82% (27/33) had IgA levels above the threshold. In relation to the trimester of infection, 83% (5/6) of the women infected in the first trimester had IgG levels above the threshold and 67% (4/6) had IgA levels above the threshold. A total of 60% (6/10) of women infected in the second trimester had IgG levels above the threshold and 91% (9/10) had IgA levels above the threshold. A total of 88% (15/17) of women infected in the third trimester had IgG levels above the threshold and 82% (14/17) had IgA levels above the threshold. (Table 3)

In total, 34 women had serum analyzed three and/or six months pp and a combined 55 serum samples were available. Of these 55 samples, 78% (43/55) had IgA and IgG above the threshold. This corresponds to 82% (28/34) of the participating women having IgG antibodies as well as IgA antibodies in their serum. 

No significant difference was seen between the trimesters of infection in relation to the frequency of breastmilk and serum antibodies in the third and sixth month pp.

### 2.3. IgA and IgG Duration and Concentration in Breastmilk and Serum

#### 2.3.1. Breastmilk

Breastmilk IgA antibodies were detectable at both three and six months pp regardless of the trimester of infection, except for the women infected in the first trimester who did not have IgA antibodies six months pp. The highest breastmilk IgA antibody concentration at three months pp was measured when the infection occurred in the second trimester, corresponding to 343 days post infection. At six months pp the highest breastmilk IgA antibody concentration was measured when the infection occurred in the third trimester, corresponding to 285 days post infection. The longest duration of breastmilk IgA antibodies above the threshold was measured when the infection occurred in the second trimester, corresponding to 350 days post infection. 

#### 2.3.2. Serum

Both serum IgG and IgA antibodies were detected at both three and six months pp regardless of the trimester of infection. In the 21 serum samples repeated at both three and six months pp, the antibody concentration remained stable or slightly increased between the third and sixth month regardless of the trimester of infection. 

The highest serum IgA antibody concentration at three months pp was measured when the infection occurred in the second trimester, corresponding to 250 days post infection. At six months pp, the highest serum IgA antibody concentration was measured when the infection occurred in the first trimester, corresponding to 400 days post infection. The longest duration of serum IgA antibodies above the threshold was measured when the infection occurred in the first trimester, corresponding to 430 days post infection (Figure 1).

The highest serum IgG antibody concentration three and six months pp was measured when the infection occurred in the second trimester, corresponding to 250- and 341 days post infection, respectively. The longest duration of serum IgG antibodies above the threshold was measured when the infection occurred in the first trimester, corresponding to 430 days post infection (Figure 1).

### 2.4. Antibody Development from Three to Six Months pp

Out of the 15 women who provided breastmilk samples at both three and six months pp, 60% had an increase or were stationary in breastmilk IgA concentration levels between the third and sixth month. In relation to the trimester of infection, 75% of the women infected in the third trimester, 62.5% of the women infected in the second trimester and 66.5% of the women infected in the first trimester showed an increase or stationary levels in IgA breastmilk concentration (Figure 2).

Out of the 21 women who provided serum samples at both three and six months pp, 71.5% had an increase or were stationary in both their serum IgA and serum IgG concentration between the third and sixth month. In relation to the trimester of infection 80% of women infected in the third trimester, 50% of the women infected in the second trimester and 100% of the women infected in the first trimester had an increase or stationary levels in IgA serum concentration. An increase or stationary serum IgG concentration levels were seen in 80% of the women infected in the third trimester, 60% of the women infected in the second trimester and 83.5% of the women infected in the first trimester (Figure 2).

### 2.5. Ratio of Breastmilk IgA and Serum IgG/IgA 

The ratio between breastmilk IgA and serum IgG/IgA was measured at three and six months and categorized by trimester of infection. No significant correlation was found between serum IgA and breastmilk IgA, nor between serum IgG and breastmilk IgA. In general, there was a tendency for breastmilk IgA antibody concentration to be lower than serum IgG and serum IgA antibody concentration. (Figure 3)

### 2.6. Admission to Pediatric Emergency Ward

In the cohort of 36 SARS-CoV-2-positive women, 38 children were born (two set of twins). In total, four children (11%) had been in contact with the hospital. One was hospitalized and three were evaluated at the pediatric emergency ward with SARS-CoV-2-related symptoms within the first six months of life. PCR testing for SARS-CoV-2 was performed on three children, all of whom tested negative. In the control group, 24 children out of 180 (13.33%) were either hospitalized or seen in the pediatric emergency ward. Four were hospitalized and 20 were evaluated at the pediatric emergency ward with above mentioned SARS-CoV-2-related symptoms as well as diarrhea, vomiting, change in eating pattern and signs of malaise. Twenty of the children were PCR-tested for SARS-CoV-2 and 18 tested negative and two tested positive. Another child tested positive but not in relation to a hospital visit. No children in the case group were premature, and therefore also none in the control group. One of the children in the control group, who were seen at the pediatric emergency ward, had a low birth weight (birthweight < 2500 g) of 2430 g. No significant difference was found in the frequency of admission to pediatric emergency ward with SARS-CoV-2-related symptoms in the first half year of life between the positive group and the control group (OR 1.23 95%CI (0.40–3.79), adjusted OR 1.23 95%CI (0.40–3.81)). 

## 3. Discussion

This study found that 27.7% of the women had breastmilk IgA levels above the threshold. In the remaining 72.3% of the women, breastmilk IgA was detectable, but below the threshold, in all samples both three and six months pp. None of the women had breastmilk IgG levels above the threshold and only 27.7% of the women had detectable breastmilk IgG. These results align with the general findings of secretory IgA as the most important and abundant component of breastmilk. IgA is responsible for 80–90% of total immunoglobulins in human milk, whereas IgG in breastmilk is found in much lower concentrations [10]. Evidence on the trimester of infection and antibody transfer in general has demonstrated a positive correlation between maternal–fetal transfer of IgG antibodies and week of gestation [11]. Studies suggest that the second trimester is the most optimal timeframe for the best protection via maternal–fetal IgG antibody transfer, due to a larger transfer window [12,13]. Examining the time of infection regarding SARS-CoV-2, the PREGCO study showed that the trimester of infection can influence the amount of maternal–fetal SARS-CoV-2 antibody transfer. Fich et al. found that 87.5%, 95.3% and 60.3% of newborns of women who tested positive in their first, second and third trimester, respectively, had higher IgG antibody levels than their mothers at delivery [6]. These findings could theoretically mean that mothers infected during the third trimester have the highest amount of antibodies when they start to breastfeed followed by mothers infected in the first and second trimester, respectively. We initially speculated that this study would see an antibody transfer to the breastmilk corresponding to these findings, but it did not. In the present study, we found the highest percentage of IgA in breastmilk in the women infected in the second trimester (36% 3MPP; 38% 6MPP), followed by the women infected in the third trimester (20% 3MPP; 28% 6MPP) and the lowest IgA transfer occurred in the women infected in the first trimester (16,6% 3MPP; 0% 6MPP). The difference might be due to the relatively small number of participants in our study. Other studies did not find an association between the antibody transfer to breastmilk and gestational age, but they were limited due to unsuccessful participation from all three trimesters [8,14]. We found the longest duration of positive serum IgG antibodies above the threshold to be 61.5 weeks from a women infected in her first trimester, and the duration may have been longer if the study had included a more extended follow-up period. This duration surpasses the majority of studies on SARS-CoV-2 (Alpha variant) IgG antibody duration observed [15,16]. The understanding of infant outcomes following maternal SARS-CoV-2 infection is still evolving, but Flaherman et al. reported different infant outcomes and found no difference in NICU admission for infants of SARS-CoV-2-infected mothers compared to infants of non-infected mothers [17]. This corresponds to the findings of this study with no significant difference in the frequency of admission to pediatric emergency ward with SARS-CoV-2-related symptoms in the first half year of life between children born to women infected with SARS-CoV-2 during pregnancy and children born to women without SARS-CoV-2 infection. We strongly recommend further investigation on the possible transfer of SARS-CoV-2 antibodies to the child when breastfed, where multiple follow-ups with blood samples from the child during the nursing period would be of great interest. 

Strengths and limitations: The participants in the study were part of the larger PREGCO cohort [6] that had minimalized selection bias, large complete medical history and a control group consisting of a participation rate of 75.1% for first trimester, 61% for second trimester and 72.5% for parturient women. All women in this study were infected with the SARS-CoV-2 Alpha variant and none of them were vaccinated, which makes the results uniform. Longitudinal collection of both the serum and breastmilk samples presents more complete findings. The limitations are the sample size and the discontinuance of sampling from both three and six months post-partum from some women. 

Conclusion: Breastmilk IgA antibodies were detectable both three and six months post-partum regardless of the trimester of infection, except for women infected in the first trimester six months pp. We did not find a correlation between serum IgA and breastmilk IgA, nor between serum IgG and breastmilk IgA. The highest IgA concentration in breastmilk was found in a sample six months pp from a woman infected in the third trimester. Serum IgA was detectable up to more than 400 days post infection, and serum IgG above the threshold was found after 61.5 weeks after infection in a woman infected in the first trimester. Our study contributes to the incomplete understanding of the timing of infection in naturally immunized mothers and its potential impact on future maternal–fetal and maternal–infant immunization.

## 4. Materials and Methods

### 4.1. Study Population

During the first pandemic wave in Denmark [18], the PREGCO cohorts were established [6]. Women from the SARS-CoV-2-positive PREGCO cohort were considered for our study if they gave birth at the Department of Obstetrics and Gynecology, Copenhagen University Hospital Hvidovre (CHH), between 4 April 2020 and 1 December 2020. Women were eligible for participation in this study if they or their newborn had a serum IgG antibody test at the time of delivery ≥10.00 AU/mL. Women were contacted via telephone, and the collection period spanned from 8 November 2020 until 1 June 2021. A few women had moved to a region not using the EPIC electronic platform for medical records, and therefore considered lost to follow-up, due to lack of access to their medical data. Given the exploratory design of this study, no formal sample size calculation has been performed (Figure 4).

### 4.2. Sample Collection

The samples were collected in the third and sixth month postpartum in the participant’s home. Four mL of venous blood was drawn from the mother and 5–10 mL of breastmilk was expressed either by hand or breast pump, following hygiene instructions provided before the visit. The expression or pumping of breastmilk was conducted during the home visit or collected by the woman prior to the visit on the same day. The collected milk was stored in a clean plastic tube provided to the participant and kept in their freezer. Milk samples were then pipetted and stored in cryotubes in a laboratory freezer at −80 °C. The maternal blood samples were centrifuged for 10 min at 23 °C at 3350 rpm/rcf, pipetted and stored in cryotubes in a laboratory freezer at −80 °C. 

### 4.3. Pediatric Emergency Admission

From the SARS-CoV-2-negative PREGCO cohort [6] a control group was matched to the 36 SARS-CoV-2-positive cases in this study. Breastfeeding was considered the primary nutrition intake for the children involved. A complete medical history and comprehensive demographic data, such as maternal age, parity, BMI, smoking, mode of delivery, gestational age at birth and obstetric complications, were available for both cases and controls (Table 2). The controls were matched 1:5 on the variables: maternal age (+/− 2 years), gestational age (+/− 3 days), maternal chronic diseases and pre-eclampsia. It was possible to match one birth complication to get five controls per case and pre-eclampsia was chosen as the most relevant obstetric complication to match with. The “ccoptimalmatch” package for R was used to match the cases with controls [19]. To calculate the difference in frequency of admission to a pediatric emergency ward with SARS-CoV-2 we used a logistic regression model. In the adjusted analysis, we included the variables: maternal age, gestational age and maternal chronic conditions. The goodness of fit was examined using the Hosmer–Lemeshow test. Data are presented with odds ratio (OR) and corresponding 95% confidence interval (CI). We checked for pediatric emergency admission with SARS-CoV-2-related symptoms or a positive SARS-CoV-2 test within the first six months of life. Symptoms included coughing, stuffy nose, fever and change in breathing pattern. Given the exploratory design of this study, no formal sample size calculation has been performed.

### 4.4. Antibody Analysis

SARS-CoV-2 spike RBD S1 domain IgA was determined using a commercially available semiquantitative enzyme immunoassay (ELISA) (Euroimmun Ag, Lübeck, Germany). Serum samples were analyzed according to instructions from the manufacturer using a fully automated ELISA instrument. The breastmilk samples were tested as serum. The samples with a ratio <0.8 AU/mL were interpreted as negative and the samples with a ratio ≥0.8 AU/mL were interpreted as positive.

SARS-CoV-2 spike RBD S1 domain IgG was determined using a commercially available semiquantitative enzyme immunoassay (ELISA) (Euroimmun Ag, Lübeck, Germany). The serum samples were analyzed according to instructions from the manufacturer using a fully automated ELISA instrument. The breastmilk samples were tested as serum. The samples with a ratio <0.8 AU/mL were interpreted as negative and the samples with a ratio ≥0.8 AU/mL were interpreted as positive. The analysis of the SARS-CoV-2 IgG antibodies used for study inclusion is described in Fich et al. [6].

### 4.5. Statistical Analysis

The categorical variables are expressed as frequencies and percentages. The continuous variables are expressed as the median and an interquartile range (IQR). A scatterplot was used to visualize IgA and IgG duration and concentration in the breastmilk and serum. A spaghettiogram was used to visualize antibody development from three to six months post-partum. A box plot was used to visualize the ratio between breastmilk IgA and serum IgG/IgA. The data were analyzed using R version 4.2.0 [20].

### 4.6. Ethics

The study was approved by the Knowledge Center for Data Protection and Compliance, the Capital Region of Denmark, and by the Scientific Ethics Committee of the Capital Region of Denmark (journal number H-20022647). All participants provided written informed consent. 

## Figures and Tables

**Figure 1 ijms-25-03269-f001:**
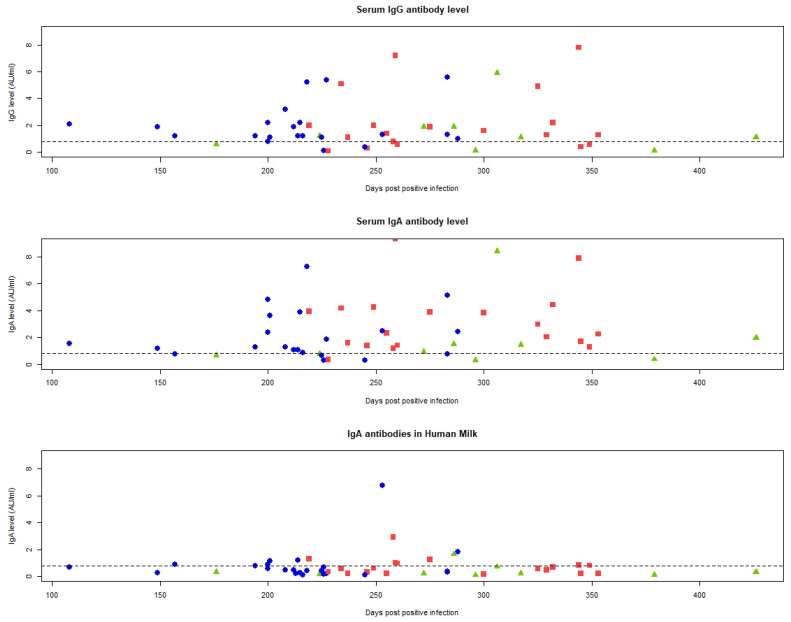
IgA and IgG duration and concentration in breastmilk and serum. Green triangle = infected in the first trimester, Red square = Infected in the second trimester, Blue dot = infected in the third trimester.

**Figure 2 ijms-25-03269-f002:**
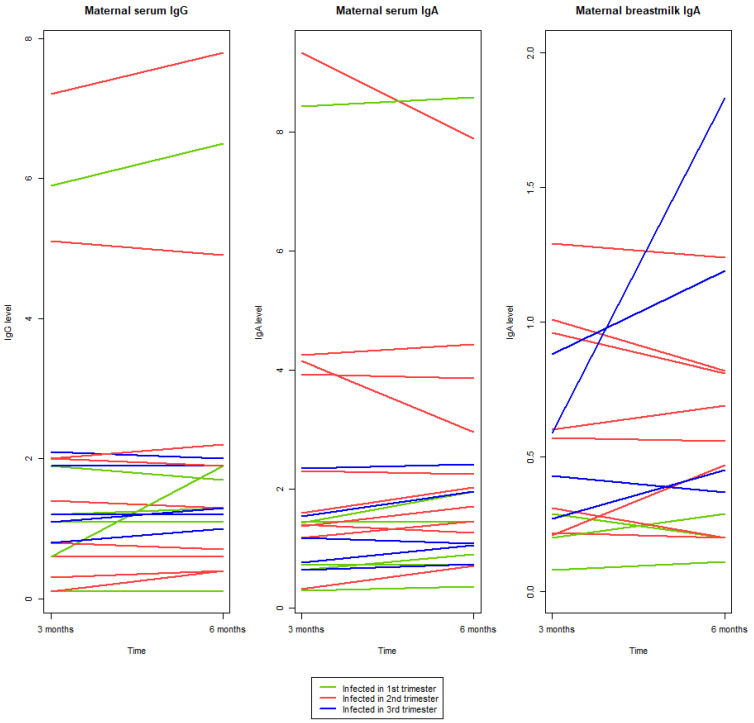
Antibody development from three to six months pp in serum and breastmilk.

**Figure 3 ijms-25-03269-f003:**
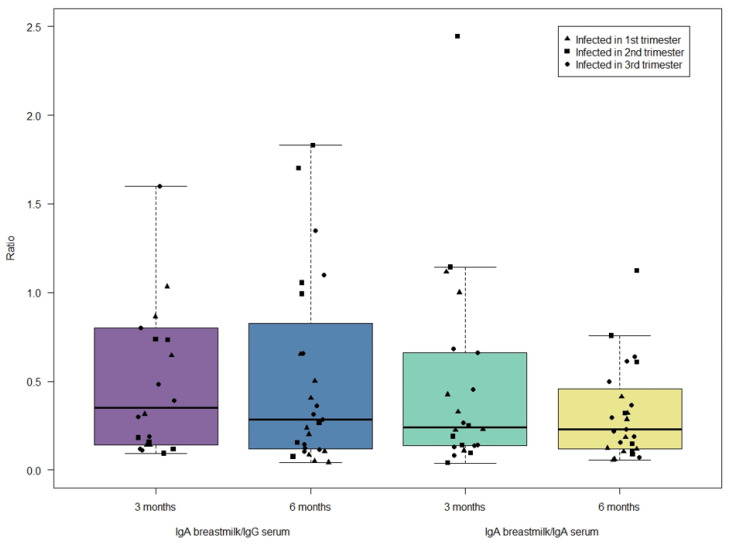
Ratio between maternal serum IgG or IgA and breastmilk IgA.

**Figure 4 ijms-25-03269-f004:**
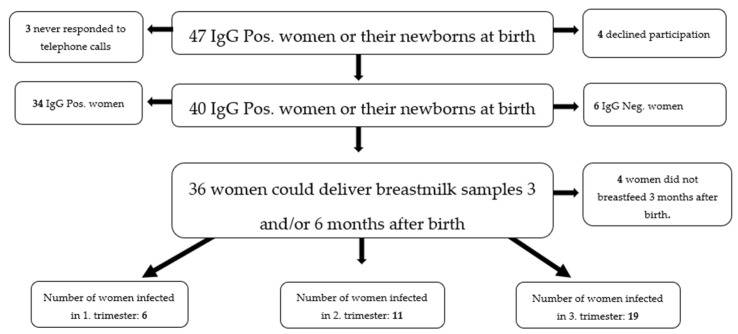
Flowchart of participants.

**Table 1 ijms-25-03269-t001:** Sample collection overview.

Study Participants	No. of Samples 3 Months after Birth	No. of Samples 6 Months after Birth
First trimester of infection:	6 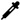	3 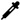
Second trimester of infection:	11 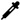	8 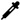
Third trimester of infection:	5 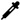	18 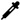

**Table 2 ijms-25-03269-t002:** Demographics.

	Cases (n = 36)		Controls (n = 180)	
Variables	n	(%)	n	(%)
Age *, years; median(IQR)	32	(29;36)	32	(29;36)
Parity				
Primipara	20	(56)	89	(49)
Multipara	16	(44)	91	(51)
Body mass index, points; median(IQR)	23	(22;25)	23	(21;27)
Smoking, yes	≤3		4	(2)
Obstetric complications, yes				
GestHT	0	(0)	4	(2)
GDM	≤3	(-)	16	(9)
Liver complications	≤3	(-)	≤3	(-)
PPROM	0	(0)	≤3	(-)
Post-partum bleeding	7	(19)	50	(28)
Acute sectio	5	(14)	23	(13)
Mode of delivery, yes				
Vaginal	27	(75)	141	(78)
Cesarian	9	(25)	39	(22)
Gestational age at birth *, days; median(IQR)	281	(277;289)	281	(277;289)
Apgar score				
At 5 min., points; median(IQR)	10	(10;10)	10	(10;10)
Less than 7 points, yes	0	(0)	≤3	(-)
Preterm birth, yes	≤3	(-)	9	(5)

Abbreviations: n = number, IQR = inter quartile range, GestHT = gestational hypertension, GDM = gestational diabetes mellitus, PPROM = preterm premature rupture of membranes, min.= minutes. * Matching variables, (-) = No appearance of the variable.

**Table 3 ijms-25-03269-t003:** IgA and IgG frequency in breastmilk and serum three and six months post-partum.

	IgG Serum		IgA Serum		IgA Breastmilk	
	3 Months	6 Months	3 Months	6 Months	3 Months	6 Months
Number	22	33	22	33	22	29
No. of samples per trimester						
First trimester	6	6	6	6	6	3
Second trimester	11	10	11	10	11	8
Third trimester	5	17	5	17	5	18
No. (%) of positive samples per trimester						
First trimester	4 (67%)	5 (83%)	3 (50%)	4 (67%)	<3	0 (0%)
Second trimester	8 (73%)	6 (60%)	10 (90%)	9 (91%)	4 (36%)	3 (38%)
Third trimester	5 (100%)	15 (88%)	3 (60%)	14 (82%)	<3	5 (28%)

## Data Availability

Data are contained within the article.

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
