# Peer review of "SARS-CoV-2 Antibodies in Breastmilk Three and Six Months Postpartum in Relation to the Trimester of Maternal SARS-CoV-2 Infection—An Exploratory Study"

_ijms, 2024, doi:10.3390/ijms25063269_

Round 1

Reviewer 1 Report

Comments and Suggestions for Authors

General comment

The paper “Original Research SARS-CoV-2 Antibodies in Breastmilk Three- and Six-Months Postpartum in Relation to the Trimester of Maternal SARS-CoV-2 Infection” is an interesting article about the prevalence, concentration and duration of IgA and IgG antibodies in breastmilk and blood serum in the third- and sixth-month post-partum in former SARS-CoV-2 infected mothers in relation to the trimester of infection. The paper is a well written and the results may be useful to future recommendation of control of SARS-CoV-2 infection of mother and newborn, however some aspects needs editorial review.

Major comments

The possible main implication of the study should be highlighted in the Abstract and Discussion (second trimester as the optimal timeframe for the best protection via maternal-fetal antibody transfer?)  and anecdotic results should be deleted (results detectable in a women...).

Specific comments

1)      Review the Abstract and add the relevant implications.

2)      Review the methods section and explain better the statistical analysis

3)      Review some results and reduce the importance of anecdotic results of only one or two patients

4)      Review the 1rst paragraph of Discussion, It is hard to follow and understand (line 187-193)

5)      The authors do not have results about the protection of antibody in cases and control; please eliminate or change the comment about it.

6)      Review the style of references.

Comments on the Quality of English Language

None

Author Response

Thank you very much to both reviewer for taking the time to review this manuscript. Please find the detailed answers and actions taken below and see the corresponding revisions in track changes in the re-submitted file. 

Best regards,

Line Fich

Reviewer 1

General comment:

The paper “Original Research SARS-CoV-2 Antibodies in Breastmilk Three- and Six-Months Postpartum in Relation to the Trimester of Maternal SARS-CoV-2 Infection” is an interesting article about the prevalence, concentration and duration of IgA and IgG antibodies in breastmilk and blood serum in the third- and sixth-month post-partum in former SARS-CoV-2 infected mothers in relation to the trimester of infection. The paper is a well written and the results may be useful to future recommendation of control of SARS-CoV-2 infection of mother and newborn, however some aspects needs editorial review.

Major comments

The possible main implication of the study should be highlighted in the Abstract and Discussion (second trimester as the optimal timeframe for the best protection via maternal-fetal antibody transfer?)  and anecdotic results should be deleted (results detectable in a women...).

Comment:

Review the Abstract and add the relevant implications.

Answer: Thank you for comment.

Action taken: Both reviewers had a comment for the abstract, and it has now been amended accordingly to the comments.

Line: 24 -43.

Comment:

Review the methods section and explain better the statistical analysis

Answer: Thank you for your input. Both reviewers have commented on this.

Action taken: Adjustments have been made and is explained in more details under reviewer 2.

Comment:

Review some results and reduce the importance of anecdotic results of only one or two patients

Answer: Thank you for pointing this out.  

Action taken: The results section with numerous anecdotic has been revised.

Line: 154-168.

Comment:

Review the 1rst paragraph of Discussion, It is hard to follow and understand (line 187-193)

Answer: Thank you for pointing this out. I can understand what you mean.

Action taken: The paragraph has been amended for clarification.  

Line: 214-220.

Comment:

The authors do not have results about the protection of antibody in cases and control; please eliminate or change the comment about it.

Answer: Thank you for your comment.

Action taken: The sentence has now been removed.

Line: 251.

Comment:

Review the style of references.

Answer: We agree with this comment.  

Action taken: The references have now been amended to match accordingly to the journals style of references.

Line: 375-432.  

Reviewer 2

General comment

SARS-CoV-2 Antibodies in Breast Milk Three and Six Months Postpartum in Relation to the Trimester of Maternal SARS-CoV-2 Infection" (ijms-2837173). This article is submitted to the Special Issue "Covid-19 and Future Pathogens" in the Section of Molecular Microbiology.

This work falls within the scope of the Special Issue and the specific section to which it is directed.

 The title is informative as it previews the content of the study.

Comment:

Regarding the abstract, it is suggested that it should be completed by clearly stating the work's objective, similar to the introduction. This includes the study design, methodology employed, and results indicating quantitative reactions in the least, unless the prevalence of immunoglobulins in blood, the main focus of the study, is emphasized. This would make the abstract more comprehensive and informative.

Answer: Thank you for pointing this out.  

Action taken: The points mentioned in your comment has been incorporated into the reconstructed abstract. Line: 24-43.

Comment:

Concerning the introduction, it is recommended to expand it. References should be cited individually, providing information from each of these works. The working hypotheses should be presented after summarizing these references, followed by the research objectives, divided into the main and secondary objectives, particularly for the case of children.

Answer: Thank you for your comment.

Action taken: The introduction has been amended accordingly to your comment.  

Line: 48-76.

Comment:

In the materials and methods section, it should be clarified whether the sample size for this study was calculated. If so, emphasizing this would strengthen the obtained results. If not, it should be explicitly stated, especially considering the seemingly small sample size.

Answer: Thank you for pointing this out. Given the exploratory design of this study no specific primary outcome is evaluated, or hypothesis tested. As such no formal sample size calculation has been performed. We have added this to the material and methods section.

Action taken: text added to line 284: “Given the exploratory design of this study no specific primary outcome is evaluated, or hypothesis tested. As such no formal sample size calculation has been performed.”

Comment:

Additional information collected from both the mother and the newborn, as indicated in Table 2, should also be included. The study design, as cases and controls are mentioned in Table 2, should be clearly defined in this section.

Answer: Thank you for your comment.

Action taken: Regarding table 2, information on full medical history and comprehensive demographics has been added. Please see

Line: 299-302.                            

Comment:

The calculated risk, assumed to be Odds Ratio (OR), needs explicit clarification.

Answer: Thank you for pointing this out. We agree with the reviewer that the calculated risk of admission to pediatric emergency should have been clarified. We have added text to the Methods section and the OR’s and corresponding confidence intervals to the Results section

Action taken: text added to line 306-311: “To calculate the difference in frequency of admission to pediatric emergency ward with SARS-CoV-2 we used a logistic regression model. In the adjusted analysis, we included the variables: maternal age, gestational age, and maternal chronic conditions. The goodness of fit was examined using Hosmer–Lemeshow test. Data are presented with Odds ratio (OR) and corresponding 95% confidence interval(CI). Text added to line: 212 “(OR 1.23 95%CI(0.40-3.79), adjusted OR 1.23 95%CI(0.40-3.81)).”

Line: 212 & line: 306-311

Comment:

Figure One, which outlines the study procedure, should be moved to the materials and methods section.

Answer: Agree.

Action taken: It has been moved to the materials and methods section.

Comment:

Table 1, containing limited content, could be condensed or discussed in the text.

Answer: Thank you for this useful suggestion. 

Action taken: We have condensed the table.

Line: 90

Comment:

Table 2 provides baseline characteristics, but it is crucial to compare cases and controls. Completing this table is essential, as it informs about the variables to consider in adjusted analyses. The same applies to Table 3; a comparison between three and six months for each of the studied immunoglobulins should be considered, as absolute values alone are somewhat informative.

Answer: Thank you for your comment. However, as we consider table 2 and 3 to be descriptive tables characterizing the sample in this study, and as we follow the STROBE guidelines that state:

“Inferential measures such as standard errors and confidence intervals should not be used to describe the variability of characteristics, and significance tests should be avoided in descriptive tables.” (w-179)

We do not agree with the reviewer. As standard errors, confidence intervals and p-values are inferential measures that try to infer something about the population, we do not consider it appropriate in a descriptive table, as we are trying to describe the sample characteristics. Furthermore, the STROBE guideline also concludes that p values are not an appropriate criterion for selecting variables to consider in adjusted analysis.

“Also, P values are not an appropriate criterion for selecting which confounders to adjust for in analysis…” (w-179)

Ref: Vandenbroucke JP, von Elm E, Altman DG, Gøtzsche PC, Mulrow CD, Pocock SJ, Poole C, Schlesselman JJ, Egger M; STROBE initiative. Strengthening the Reporting of Observational Studies in Epidemiology (STROBE): explanation and elaboration. Ann Intern Med. 2007 Oct 16;147(8):W163-94. doi: 10.7326/0003-4819-147-8-200710160-00010-w1. PMID: 17938389.

Action taken: None

Comment:

Figure Four lacks a title and should have one stated at the top, explaining the information it contains.

Answer: Thank you for pointing this out.

Action taken: The title has been adapted.

Line: 190

Comment:

In the section on pediatric admission of the studied children, the small sample size should be justified. Additionally, clarification is needed on whether breastfeeding was exclusive and if the children had any vulnerabilities such as prematurity or low weight that could influence their subjectivity or alter their immune response.

Answer: We agree with this comment.

Action taken: Sample size is now justified line: 313-315, breastfeeding clarified line: 398-399, and children vulnerabilities statement incorporated line: 207-209. 

Reviewer 2 Report

Comments and Suggestions for Authors

"SARS-CoV-2 Antibodies in Breast Milk Three and Six Months Postpartum in Relation to the Trimester of Maternal SARS-CoV-2 Infection" (ijms-2837173). This article is submitted to the Special Issue "Covid-19 and Future Pathogens" in the Section of Molecular Microbiology.

This work falls within the scope of the Special Issue and the specific section to which it is directed.

The title is informative as it previews the content of the study.

Regarding the abstract, it is suggested that it should be completed by clearly stating the work's objective, similar to the introduction. This includes the study design, methodology employed, and results indicating quantitative reactions in the least, unless the prevalence of immunoglobulins in blood, the main focus of the study, is emphasized. This would make the abstract more comprehensive and informative.

Concerning the introduction, it is recommended to expand it. References should be cited individually, providing information from each of these works. The working hypotheses should be presented after summarizing these references, followed by the research objectives, divided into the main and secondary objectives, particularly for the case of children.

In the materials and methods section, it should be clarified whether the sample size for this study was calculated. If so, emphasizing this would strengthen the obtained results. If not, it should be explicitly stated, especially considering the seemingly small sample size. Additional information collected from both the mother and the newborn, as indicated in Table 2, should also be included. The study design, as cases and controls are mentioned in Table 2, should be clearly defined in this section. The calculated risk, assumed to be Odds Ratio (OR), needs explicit clarification.

Figure One, which outlines the study procedure, should be moved to the materials and methods section.

Table 1, containing limited content, could be condensed or discussed in the text.

Table 2 provides baseline characteristics, but it is crucial to compare cases and controls. Completing this table is essential, as it informs about the variables to consider in adjusted analyses. The same applies to Table 3; a comparison between three and six months for each of the studied immunoglobulins should be considered, as absolute values alone are somewhat informative.

Figure Four lacks a title and should have one stated at the top, explaining the information it contains.

In the section on pediatric admission of the studied children, the small sample size should be justified. Additionally, clarification is needed on whether breastfeeding was exclusive, and if the children had any vulnerabilities such as prematurity or low weight that could influence their subjectivity or alter their immune response.

Author Response

(The authors gave the same response as above.)

Round 2

Reviewer 1 Report

Comments and Suggestions for Authors Thanks for the new version of the article and the changes made

Author Response

Thank you for taking your time to review the manuscript to help make it a better reading experience for the reader, and thank you for your comment: 

"Thanks for the new version of the article and the changes made"

Best regards,

Line Fich

Reviewer 2 Report

Comments and Suggestions for Authors

I have carefully reviewed the revised version of the manuscript entitled "SARS-CoV-2 Antibodies in Breast Milk Three and Six Months Postpartum in relation to the Trimester of Maternal SARS-CoV-2 Infection" (ijms-2837173), along with the authors' response to the suggestions made for improving this manuscript.

I appreciate the authors' clarifications regarding the incorporation of more precise information into the manuscript. However, while structuring the objectives, the authors initially aim to understand the prevalence, concentration, and duration of IgA and IgG antibodies in breast milk and blood serum in the third and sixth months postpartum in mothers previously infected with SARS-CoV-2. The objectives presented in the introduction do not align with what is stated in lines 305-307.

The authors have not performed any sample size calculations for these objectives. In the materials and methods section, it is mentioned in Figure 4 that the sample size of women who continue until 6 months with three samples is only 36. This sample size appears to be notably small, allowing for the presentation of antibody frequency in a sample at most, but not the prevalence at the population level.

The authors do not define the design used, though they base it on a cohort. If it is indeed a subcohort, they should present the relative risk instead of the odds ratio (OR). Alternatively, is it a nested case-control within a cohort? For presenting prevalence, a cross-sectional design would be more appropriate. These aspects need clarification. Moreover, if the sample size justification for the stated objectives cannot be substantiated, considering this as a pilot study should be contemplated.

I acknowledge the rationale provided for not comparing the descriptive tables. However, the epidemiological design and specific objectives of the study remain unclear to me, hindering a more thorough assessment of the work.

Author Response

Thank you for taking the time to review. Please see comments and answers in the the attachment.

Round 3

Reviewer 2 Report

Comments and Suggestions for Authors

I have revisited the article titled "SARS-CoV-2 Antibodies in Breast Milk Three and Six Months Postpartum in relation to the Trimester of Maternal SARS-CoV-2 Infection" (ijms-2837173), along with the authors' responses to the provided comments.

I have observed that the authors have clarified the methodology employed, explicitly stating the utilization of two designs – a cohort and a case-control design. Additionally, they have justified not conducting a sample size calculation, attributing it to the exploratory nature of the study. However, I believe that if it is indeed an exploratory study, this should be reflected in the title of the work.
